# Hierarchical Porous SiO_2_ Cryogel via Sol-Gel Process

**DOI:** 10.3390/gels8120808

**Published:** 2022-12-09

**Authors:** Marius Horváth, Katalin Sinkó

**Affiliations:** Institute of Chemistry, Eötvös Lóránd University, H-1117 Budapest, Hungary

**Keywords:** porous silica cryogel, sol-gel chemistry, freeze-drying

## Abstract

The aim of this research work was to develop a new, low-cost and low-energy-consuming preparation route for highly porous silica systems. The precursor gel systems were synthesized by sol-gel chemistry. The starting materials were TEOS and water glass in the sol-gel syntheses. The effect of the chemical composition, the catalysis, the pH, and the additives were investigated on the structure and porosity of the cryogels. The gel systems were treated by freeze-drying process to obtain porous cryogel silica products. The cryogel systems possess hierarchical structures. The conditions of the freeze-drying process were also studied to increase the porosity. Small angle X-ray measurements, scanning electron microscope technique, and infrared spectroscopy were applied for the investigation of gel and cryogel systems.

## 1. Introduction

Recently, there has been increasing interest towards materials with hierarchical porous systems for thermal insulation applications [1]. Silica cryogels belong to these types of materials, with a typical porosity of around 40–70%, providing a cost-effective alternative to aerogels. These materials are special derivatives of gel systems, in which the solvent encapsulated in the three-dimensional network is substituted with air during a process called freeze-drying, resulting in a porous structure with a large specific surface area and low thermal conductivity [2].

The usage of sol-gel synthesis enables the formation of a continuous 3D network, from various organic or inorganic precursors, at temperatures below 100 °C. Frequently used precursors include metal alkoxides, such as tetraethyl orthosilicate or tetramethyl orthosilicate, and inorganic silicate compounds, such as Na_2_SiO_3_ (water glass) [3]. Apart from the quality of precursors, the solvent used during the gelation also effects the gel quality. In terms of solvents, two main types of gel can be distinguished: alcohols, resulting in alcogels, and water solvent in hydrogels.

The gelation process is based on the hydrolysis and condensation reactions [4,5]. The structure of the 3D gel network is highly dependent on the ratio of hydrolysis and condensation reactions, offering excellent capabilities of controlling the fine structure of the gel [6]. Multiple parameters, such as the quality of the precursor and solvent, the pH of the system, the type of catalyst, the temperature of the synthesis, and the concentration ratio of the precursor and solvent/water can affect the gel network [7,8,9,10].

In a typical study on thermal insulating SiO_2_ materials, cryogels were produced from the alcoholic (ethanol) solution of TEOS with acidic catalysis (HCl) and a molar ratio of 1/10/1 of Si:ethanol:HCl. The study found an average particle size of 40–100 nm, with an average pore size of 1–10 nm [11].

During a study on the effectiveness of different drying techniques on the gel structure [12], waterglass-based cryogels were prepared with water solvent and hydrochloric acid as the catalyst. The cryogel could be characterized by an average pore size of 2.2 nm, which compared to the mesoporous structure of the TEOS based cryogel [11] shows a nanoporous structure.

The gel systems must be dried under special conditions to obtain porous products. The solvent of alco- or hydrogels is removed by the drying process at 80 °C under atmospheric pressure, producing xerogels. The other possible parameters for drying may be the supercritical pressure for aerogels, and the vacuum yielding cryogels. Freeze-drying is also known as lyophilisation. Pore size, pore volume, and pore morphology are dependent on variables such as freeze temperature, freezing rate, solution concentration, the nature of the solvent and solute, and the control of the freeze direction [13,14]. During the freezing process the pore liquid is frozen, and the resulting solid/ice is sublimated under vacuum, leading to the formation of porous structures.

The aim of this research work was to develop a new, low-cost and low-energy-consuming preparation route for highly porous silica systems. The gel syntheses were performed by sol-gel chemistry, and the gels were dried under freeze-drying conditions.

## 2. Results and Discussion

### 2.1. Gel Synthesis and Freeze-Drying Process

Two types of Si-precursors were provided as initial materials in the syntheses, an organic compound: tetraethoxy silane (TEOS, Si(OC_2_H_5_)_4_) and an inorganic compound: water glass (Na_2_SiO_3_). The syntheses were based on sol-gel chemistry. A 3D network is built by inorganic polymerisation, generally at <100 °C. The first step of polymerisation is the hydrolysis:≡Si−_(al/aq)_ + H_2_O ⇆ HO-Si≡_(al/aq)_ + H^+^
(1)

In the case of TEOS, ethanol was the solvent (Si_(al)_), and water was needed for the dissolution of water glass (Si_(aq)_). The second step of polymerisation is the condensation forming the 3D gel network.
≡Si-OH_(al/aq)_ + HO-Si≡_(al/aq)_ ⇆ ≡Si**−O−**Si≡ + H_2_O(2)

Starting from TEOS, alcogels are formed; then, the hydrogels are organised from the water glass. The 3D gel structures can be efficiently applied to prepare highly porous systems.

Out of the possible drying processes, freeze-drying was chosen in our experiment owing to its highly porous products and far lower cost and energy consumption. The freezing rate strongly depends on the porosity and pore size of cryogels. The faster the freezing, the higher the porosity and the smaller the pores. Regarding the required higher freezing, the freezing step was performed in liquid N_2_. The lyophilisation must be carried out at around −80 °C in order to obtain monolith particles. At higher temperature, powders can be obtained rather than monolith particles.

#### 2.1.1. Gel Synthesis from TEOS

For gelation, the alcoholic solution of TEOS must be acidic or basic catalysed. The nitric acid proved to be the most suitable acid. The main part of nitrate ions can escape during the gelation and heat treatment. In the case of using HCl or H_2_SO_4_, the anions remain in the system after heating at 500 °C, and the acetic acid is too weak for effective gelation.

Gelation lasted 50–60 h at 80 °C if the molar ratio of TEOS/ethanol/HNO_3_ was the frequently published 1:10:1. By changing this molar ratio to 1:4:3, the gelation time reduced from 50–60 h to 2 h. This procedure completed with lyophilisation resulted in a low porosity (<40%) due to the ethanol content. The ethanol featured a limited sublimation ability. If half of the ethanol volume was replaced by water in the half time of gelation, the porosity could be drastically increased from ~30% to ~70% (Figure 1 and Table 1). The size of the macropores increased from 4–5 µm to 5–15 µm. Owing to the hierarchical pore system, the size of the mesopore in the wall of macropores was modified from 40–400 nm to 60–330 nm. The wall of macropores became thinner, from 1–2 µm to 100–200 nm.

The addition of 2 M of NH_3_ to the solution resulted in an immediate formation of a sol, which was lyophilized to obtain a white powder. A colloidal 3D structure that was randomly connected formed in a basic condition instead of the polymer 3D network. The powders were *uniform* in size at 0.6–0.7 µm. The cryogels featured a more compact structure, and the total pore volume was 38–45% (Table 1).

The use of citric acid beside nitric acid had no strong effect on gelation, but the cryogels showed a high level of shrinkage (Figure 1). Their porosity was generally less than 50%, and the size of the macropores was 1–2 µm. However, instead of macropores, the meso- and nanopores were dominated; their size changed from 40 nm to 150 nm. 

Table 1 represents the effect of the slow freezing rate on the porosity and pore size. (The row is denoted by *) The porosity was drastically reduced by this slow rate.

#### 2.1.2. Gel Synthesis from Water Glass (wgl)

The basicity of the water glass solution must be reduced from pH = 12 to below 10 in order to obtain a 3D gel structure. The gel structure is given by the polymerised results of various silicic acids (e.g., metasilicic acid, H_2_SiO_3_; orthosilicic acid, H_4_SiO_4_; disilicic acid, H_2_Si_2_O_5_). The general formula of the polymerised silicic acids is [SiO_x_(OH)_4−2x_]_n_. The following reactions can be taken into account by the reduction in pH starting from Na_2_SiO_3_ [14].
H_2_SiO_4_^2−^_(aq)_ + H^+^ → H_3_SiO_4_^−^_(aq)_ + H^+^ → H_4_SiO_4(aq)_ + H^+^ → H_5_SiO_4_^+^_(aq)_


H_2_SiO_4_^2−^ and H_3_SiO_4_^−^ silicic acid units are dominant in strong basic (pH = 9–10) solutions. H_3_SiO_4_^−^ and H_4_SiO_4_ species are significant in weak basic or neutral (pH = 6–8) solutions. A typical condensation reaction in basic solutions [12] is:H_3_SiO_4_^−^_(aq)_ + H_4_SiO_4(aq)_ ⇆ (OH)_3_**Si−O−Si**(OH)_3(aq)_ + OH^−^

The dimer species are polymerised further to trimer with a H_3_SiO_4_^−^ unit, and finally, to form a rather dense 3D network. In strong acidic (pH < 6) solutions, H_4_SiO_4_ and H_5_SiO_4_^+^ units form different dimers, and finally, loose 3D structures. The typical condensation reaction in acidic solutions [15] is:H_4_SiO_4(aq)_ + H_5_SiO_4_^+^_(aq)_ ⇆ (OH)_3_**Si−O−Si**(OH)_3(aq)_ + OH^−^ + H_2_O 

The first experiences aimed to determine the ideal acid application. The use of H_2_SO_4_ or HClO_4_ resulted in precipitates. The acetic acid was too weak to attain the pH < 6. The use of HCl or HNO_3_ obtained optically clear, homogeneous gel systems. Further experiments concentrated on the application of HNO_3_ due to the much easier escape of nitrate anions than chloride. The main part of nitrate anions could be decomposed and evaporated during the gelation and drying process at 80 °C. The catalysis with HNO_3_ yielded a more homogeneous and porous gel structure than HCl (Figure 2 and Table 2). The decrease in pH can be realised by cation exchange. A cation exchange resin is capable of replacing the Na^+^ ions in the water glass solution for the H^+^ ions. The use of ion exchange considerably increased the preparation time. Even the gelation time could not be reduced below 20 h because a strongly diluted solution must be used for the ion exchange resin column; otherwise, the gelation on the column cannot be avoided. The cryogels resulting from the ion exchange possessed medium porosity, which was solely provided by the macropores.

The other condition, which requires optimalisation, is the pH value. In order to obtain homogeneous, optically clear, stable gel systems, the pH value must be reduced below 6. The pH can easily be regulated by the amount of acid. However, not only must the volume of the given acid solution be increased, but also its concentration. A large amount of water strongly hinders gelation. Table 2 summarises the effects of pH on gelation time and porous structure. The porous structure of the cryogels prepared in acidic medium was hierarchical, and it was formed from the macropores and nanopores (20–300 nm) in the wall of macropores (Table 2). 

Figure 3 represents the SEM images of the cryogel samples prepared in an acidic medium of various pH. The double SEM pictures of the given pH introduce the homogeneity of the morphology. The cryogel sample of pH 1 possessed the most homogenous structure and the highest porosity. 

### 2.2. ^29^Si MAS NMR Measurements

The **^29^Si MAS NMR** chemical shifts are characteristic of the Q^n^ species of Si, where *n* represents the number of bridging oxygens to other silicon atoms. In a glassy silica sample, the total range of Q^4^, Si(OSi)_4_ signal was found from −100 to −125 ppm; the Si NMR maximum at −110 ppm corresponded to a mean Si-O-Si angle of 146° [15]. The Si NMR peak of bulk amorphous silica particles appeared at around −110 ppm [16,17]. Each non-bridging oxygen atom induced a chemical shift of between 5 and 10 ppm with respect to the Si NMR peak position of Q^4^ silica tetrahedron [15,18]. All the modifier ions (e.g., alkaline ions) produced non-bridging oxygens (−O^−^). The replacement of O-Si group with hydroxyl group resulted in a shift of −10 ppm in the NMR spectra [18,19]. The chemical shift at around −100 ppm can be attributed to Q^3^, e.g., Si(OSi)_3_(OH) [18,19,20] or Si(OSi)_3_O^−^ [16,21]. The Si NMR signals of Q^2^ units at −90–−95 ppm and Q^1^ at −80–−85 ppm can be attached to the presence of Si(OSi)_1–2_(OH)_2–3_ [18,20] or Si(OSi)_1–2_(O^−^)_2–3_, respectively [16,22]. 

The Q^1^, Si(OSi)_1_ peak was missing in the spectra of all silica samples due to the continuous 3D gel network. The peak positions show a good correspondence in the spectra of all three silica samples (Figure 4 and Table 3). An exact 10 ppm difference can be seen between the peak positions in every case. This difference was caused by the Si–O^−^H/Na groups [18]. Significant distinction can be observed in the intensity of the peaks (Table 3). The Q^4^, Si(OSi)_4_ units were more dominant in the silica gel of TEOS than in the water glass gel. Many Q^4^ units mean a more compact structure for the elementary units, and this sample of TEOS was prepared in pure ethanol. The NMR spectra of the cryogels produced from TEOS in ethanol–water solvent or from wgl were very similar. In both spectra (Figure 4), the Q^3^ species were dominant (~60%) due to their loose network. The loose network requires many terminal groups; Si-OH in the 3D silica network was derived from TEOS and Si-O-Na in the network of water glass.

The estimated ratio of OH groups to Si atoms was 6:10 in both the gel and cryogel samples obtained from TEOS. (The peak intensity of Q^2^ and Q^3^ was the basic catalyst of estimation). The unchanged ratio proves that the number of OH groups did not decrease during the freeze-drying and drying process, thereby retaining the loose structures.

### 2.3. FTIR Measurements 

The signal of Q^0^ (Si(OH)_4_) was missing in the range of 850–890 cm^−1^ of every spectrum, supporting the Si NMR results. 

The absorption bands in the range of 3000–3800 cm^−1^ can be attributed to the stretching vibration (ν) of Si-OH (3300–3400 cm^−1^) and some residual water molecules (3400–3600 cm^−1^) (Figure 5). Regarding the intensity of these signals, most of the Si-OH groups were in the cryogel of TEOS. This is also verified by the intensity of the band (νSi-OH) at 952 cm^−1^. The peak of 952 cm^−1^ reduced and appeared at 961 cm^−1^ (νSi-OH/^−^) in the spectra of cryogel from water glass due to the Na^+^ ions [21]. 

The most intensive FTIR signals belonged to the asymmetric stretching vibration of the Si–O–Si bonds in all spectra. This band can be mostly associated with Q^3^ and Q^4^. The Si–O–Si bonds appeared at 1077 cm^−1^ in the spectrum of cryogel from TEOS. This peak shifted to lower wavenumbers in the spectra of the wgl cryogels by the presence of Na^+^. The Na^+^ decreased the strength of the Si–O–Si bonds [21,23]. 

The large band at 1364–1368 cm^−1^ in the spectra of the cryogels from wgl can be strongly attributed to the presence of the Na^+^ ion. One source of the band is the vibration of Si–O in a slightly ordered Na silicate [21]. The other source is the nitrate anion (ν_as_). The cryogels were dried at 100 °C, and a small part of the nitrate ions remained in the gel. The nitrate anions were better preserved in the wgl cryogels beside Na^+^ ions than in the cryogels of TEOS. In the system of TEOS, the nitrate anions can adopt H^+^ ions to form HNO_3_, and HNO_3_ molecules can more easily escape from less polar solution. Some difference could be observed in the intensity of the peak at 961 cm^−1^ between the spectra of the water glass cryogels, in the cryogel prepared in pH 1 medium, owing to the greater number of Si-OH groups (Figure 5). 

### 2.4. SAXS Measurements 

In the case of the gels, small angle X-ray scattering (SAXS) curves were interpreted in the frame of the fractal theory. The SAXS investigation proves that the silica alcogel prepared from TEOS in acidic medium can be characterised by a fractal structure in a wide size range. The slope (µ) of the SAXS curve in the log-log plot was −2.16, and this value belongs to a loose fractal structure with 2.16 dimension. The elementary units, which build up the fractal structure, also possessed a fractal character. The structure of the waterglass hydrogel also formed a fractal skeleton; however, this structure was somewhat more compact with a dimension of 2.69. In contrast to the fractal-like elementary units of the TEOS-derived gel, the building units in the hydrogel of wgl were compact (see the SAXS curve of gel from wgl in Figure 6); their surface fractal dimension was 6 − 3.55 = 2.45, which indicates compact particles with a rough surface. The simple interpretation of the fractal theory for the Porod-region was not carried out for the SAXS data of the cryogels owing to the “S” shape of the curve. The SAXS curve of the cryogel could be evaluated by Freltoft approximation results in unit sizes of 40–50 nm in the cryogel prepared from TEOS [24]. The dimension (slope) obtained by the Freltoft evaluation was 2.75. This dimension value indicates that the silica skeleton retained its fractal—but somewhat more compact—character during freeze-drying. 

The looser, more porous structure of alcogel from TEOS should result in larger porosity; however, the cryogel derived from wgl had a 70–78% total pore content, and the cryogel of TEOS had only 30–40%. If half the amount of ethanol is replaced by water before lyophilisation, the porosity increases to 70–72%. These results verify that not only does the looseness of the precursor gel affect the final cryogel porosity, but the quality of the solvent more considerably influences it. 

## 3. Conclusions

The aim of this research work was to develop a new, low-cost and low-energy-consuming preparation route for highly porous silica systems. The precursor gel systems were synthesized by sol-gel chemistry. The sol-gel syntheses proceeded from TEOS and water glass (wgl). The gel systems were treated by a freeze-drying process to obtain porous cryogel silica products. The cryogel systems possess hierarchical structures. In this study, the macropores were 5–15 µm in the TEOS-derived cryogels and 3–6 µm in the cryogels of wgl, and the mesopores in the wall of macropores were 60–330 nm (TEOS) and 20–60 nm (wgl). 

In the gelation of TEOS, the acidic catalyst is much more suitable than the basic one. The acidic conditions result in a 3D gel network, which is favourable to produce a highly porous system by freeze-drying. The nitric acid proved to be the most suitable acid. The main part of the nitrate ions can escape during the gelation and heat treatment. Using HCl or H_2_SO_4_, all anions remain in the system, while the acetic acid is too weak for effective gelation. 

The molar ratio, water, and acid catalyst content have a strong effect on the gelation time, besides the density of the structure. The gelation time can be reduced from 50–60 h to 2 h by an efficient molar ratio. 

The gel network derived from TEOS possesses a loose fractal character. The silica skeleton retains its fractal—but more compact—feature during freeze-drying. However, the cryogels of TEOS have a low porosity (30–40%) owing to the limited sublimation ability of ethanol. The change of half the volume of ethanol by water before freeze-drying leads to the drastic increase in porosity from ~30% to ~70%.

In the synthesis of silica gel from water glass (wgl), the pH value must be reduced to below 6 to obtain homogeneous, optically clear, stable gel systems. The large amount of water strongly hinders gelation, thus, a more concentrated acid solution (4–8 M for H^+^) is worth using. The HNO_3_ yields the most homogeneous and most porous gel structure in pH 1 medium. 

The structure of hydrogel from wgl is somewhat more compact (its mass fractal dimension is 2.69); the mass fractal dimension alcogel skeleton of TEOS is 2.19. The looser, more porous structure of alcogel should result in larger porosity. However, the cryogel derived from wgl has a 70–78% total pore content, while the cryogel prepared from TEOS in ethanol without water has only 30–40% porosity. These results verify that not only does the density of precursor gel structures have an effect on the final cryogel porosity, but the quality of the solvent more considerably influences it. The water solvent is ideal for freeze-drying. The other conditions of lyophilisation, i.e., the rate and the temperature of freezing possesses, also have a significant effect on the porosity. The application of a fast freezing rate and low temperature (−80 °C) are favourable. 

If the main aim is to produce highly porous silica products, the water glass is worth using. The gelation of wgl can be carried out by a shorter gelation time, which is a simpler, lower cost, and lower energy-consuming procedure. Wgl is able to provide the cryogel systems with a highly porous and homogeneous network. The advantage of TEOS application may be to tailor a special structure for a functional material. 

## 4. Experimental Procedure

### 4.1. Synthesis

#### 4.1.1. Gel Syntheses from TEOS

Tetraethoxy silane, Si(OC_2_H_5_)_4_ (TEOS, Sigma-Aldrich, St. Louis, MO, USA, 98%), was dissolved in ethanol and treated at 80 °C for 50–60 h in the presence of nitric acid (VWR, 65.6%) with a concentration of 2 mol/dm^3^. The typical published molar ratio of the starting composition is 1:10:1 of TEOS/ethanol/HNO_3_; however, the gelation process was lengthy when using this molar ratio. If the molar ratio of TEOS/ethanol/HNO_3_ was modified from 1:10:1 to 1:4:3, gelation occurred much quicker, reducing from 50–60 h to 2 h. The quantity characterisation of the synthesis can be demonstrated by 50 g of TEOS as the starting material. After an aging process of 2 days at room temperature, the obtained gels were then frozen with liquid nitrogen and freeze-dried for 48 h under vacuum (0.003 mbar, −60 °C, Christ, Alpha 2–4 LD plus cryostat). The cryogel derived from TEOS can be characterized by a low porosity (<25–30%). The final heat treatment was carried out at 500 °C.

In order to improve the porosity of the cryogel, half the amount of ethanol was removed by vacuum distillation. The evaporated solvent was replaced by water, and the system was kept at room temperature until gelation, which took place in one hour. After aging, the gels were freeze-dried for 48 h under vacuum. 

Three types of additives (NH_3_, VWR 96%; citric acid, Sigma-Aldrich; and propylene oxide, Sigma-Aldrich) were also investigated. The molar ratio of ammonia was 0.02 to Si; in the case of citric acid it was 0.002; and the propylene oxide was used in a 0.057 ratio to Si. 

#### 4.1.2. Gel Syntheses from Water Glass

Water glass, Na_2_SiO_3(aq)_ (VWR, aqueous solution of 36.7 *w*/*w*%), was diluted to obtain a solution of 10 *w*/*w*%. By acidic catalysis, a gel formed immediately or in a couple of hours depending on the volume of water. After gelation, the gels were kept at room temperature for the aging process for 2 days at room temperature. The gels were frozen in liquid nitrogen and the solvent crystals were sublimated in a freeze-dryer for 72 h in a pressure of 0.001 mbar (−60 °C, Christ, Alpha 2-4 LD plus cryostat). The final heat treatment was also carried out at 500 °C.

Various acids were tested in this study: nitric acid (2 M HNO_3_, VWR), perchloric acid (2 M HClO_4_, VWR), and hydrochloric acid (2 M HCl, VWR). The molar ratio of the acids was 0.036 to Si in every case. The quantity characterisation of the preparation can be demonstrated by 50 g of 10 *w*/*w*% Na_2_SiO_3_ solution as the starting material.

For gelation, the pH of the water glass solution must be reduced from 12 to below 6. The reduction occurs by effect of acidic catalysis or by means of ion exchange. The ion exchange was carried out using a cation exchange resin, which is capable of replacing the Na^+^ ions with H^+^ ions. The ion exchange was also carried out on a diluted (10 *w*/*w*%) sodium silicate solution. The ion exchange column was set up to form a droplet every 1–2 s. This way, the pH of the system reached 6, and gelation occurred in 20 h.

The effect of pH value from 1 to 11 was also investigated. In this series, 2 and 8 mol/dm^3^ (M) of HNO_3_ was applied as an acidic catalyst. Gelation lasted 0–20 h. 

### 4.2. Characterization Methods

^29^*Si Magic Angle Spinning (MAS) Nuclear Magnetic Resonance (NMR)* spectra were acquired at a 10 kHz sample rotation speed on a BRUKER DRX 500 (magnetic field 11.744 T) spectrometer (Bruker, Yokohama, Japan) with a 4 mm MAS probe, operating at a Larmor frequency of 99.36 MHz at room temperature. A single pulse sequence with a flip angle of approximately 30° (3 μs) was applied with a relaxation delay of 45 s. In total, 1000–2000 scans were detected for each spectrum. The spectrum width was 176 ppm. The chemical shift was referenced to an external TMS (tetramethylsilane, (CH_3_)_4_Si). In silica-based systems, the silicon atoms occupy the centre of an oxygen tetrahedron. The silicate units are usually represented by Q^n^, which refers to a silicon with *n* bridging oxygens to other silicon atoms. The different Q species have resonances in separate regions of the ^29^Si MAS NMR spectrum [19,20,21,22,23,24]. The deconvolution of the NMR spectra was performed to obtain Q^n^ distribution, using a regular assumption of Lorentzian functions for each type of Q^n^ units.

*Attenuated Total Reflectance (ATR) Fourier Transform Infrared (FTIR)* measurements were monitored on a Brucker IFS 55 instrument with a diamond ATR head (PIKE technology). All infrared spectra were collected over the range of wavenumber 4000–550 cm^−1^. 

The particle size and morphology were studied by a FEI Quanta 3D FEG *scanning electron microscope* (SEM, Hillsboro, OR, USA). The SEM images were prepared by the Everhart–Thornley secondary electron detector (ETD) with an ultimate resolution of 1–2 nm. Since the conductance of the particles investigated was high enough to remove the electric charge accumulated on the surface, the SEM images were performed in a high vacuum without any coverage on the specimen surface. For the best SEM visibility, the particles were deposited on a HOPG (graphite) substrate surface. SEM combined with energy dispersive X-ray spectroscopy (EDX) was mainly applied for the spatially resolved chemical analysis of monolith samples. The pore sizes and the pore volumes were determined by means of SEM images using Amira 5.2.2 software.

For the *small angle X-ray scattering (SAXS) measurements*, we used a 12 kW X-ray generator and a pinhole X-ray camera with variable distances from the sample to the two-dimensional detector (20.5–98.5 cm). The two-dimensional spectra were corrected for background (pinhole and covering foil) and then radially averaged to obtain the scattered intensity, I(q), as a function of the scattering vector. The evaluation of the SAXS curves was carried out by means of the slope of the curves in the Porod-region in the cases of the gels. The slope of the curve provides the dimension of the structure. The curve of the cryogel was interpreted by the Freltoft expression [25]. For aero- and cryogels, Freltoft proposed the following expression:$$S(q)=1+\frac{C\cdot \xi^{D}}{q\xi }\frac{1}{{\left(1+q^{2}\xi^{2}\right)}^{(D-1)/2}}\Gamma (D-1)\cdot \sin\left(\left(D-1)\cdot \arctan(q\xi \right)\right)$$
where *S*(*q*) is the structure factor, *C* is a constant, r is the radius of the particles, *D* indicates the fractal dimension, and *ξ* is the effective cut-off length describing the decay of the fractal-like correlations (due to the finite size of aggregates or their overlapping) [23].

## Figures and Tables

**Figure 1 gels-08-00808-f001:**
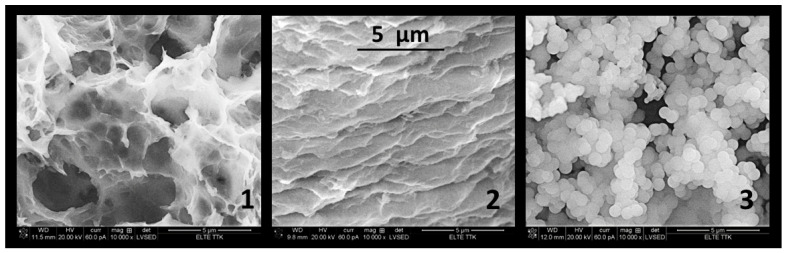
SEM images of silica cryogels from TEOS. Silica cryogel 1 is prepared from optimized composition (1:4:3) of TEOS, ethanol/water, and HNO_3_. Silica cryogel 2 is catalysed by citric acid; cryogel 3 by ammonia. The magnification is 10,000× in every image.

**Figure 2 gels-08-00808-f002:**
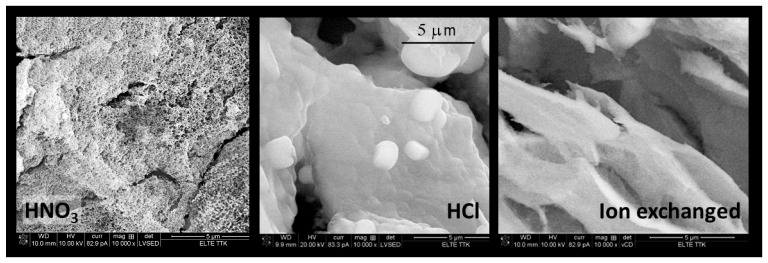
SEM images of cryogel prepared from water glass. The pH is regulated by HNO_3_, HCl, and ion exchange. Magnification is 10,000×.

**Figure 3 gels-08-00808-f003:**
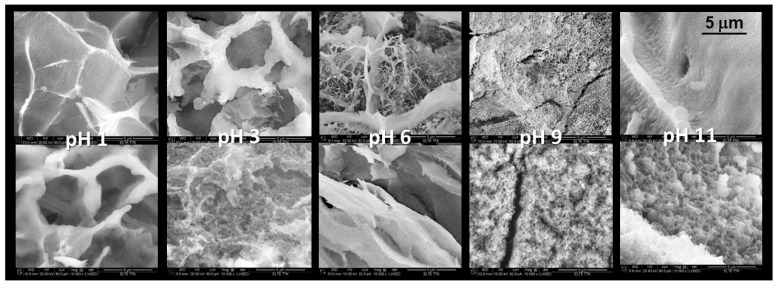
SEM images of cryogel prepared from water glass prepared in different pH. Magnification is 10,000× in every image.

**Figure 4 gels-08-00808-f004:**
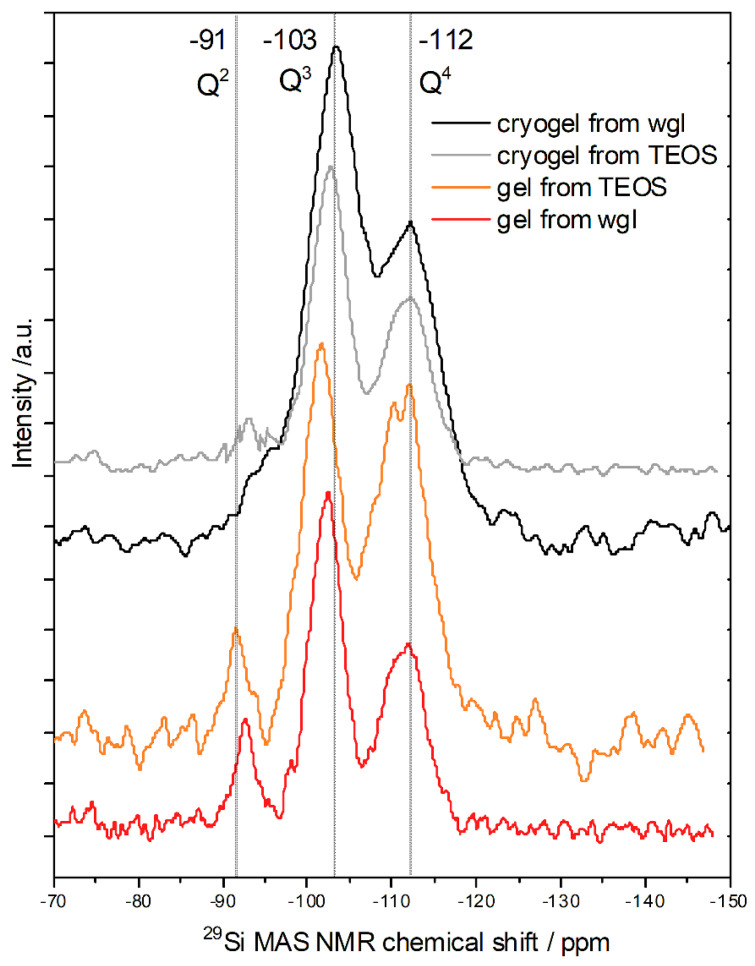
^29^Si MAS NMR spectra of cryogels (from water glass or TEOS), alcogel (from TEOS), and hydrogel (from water glass, wgl).

**Figure 5 gels-08-00808-f005:**
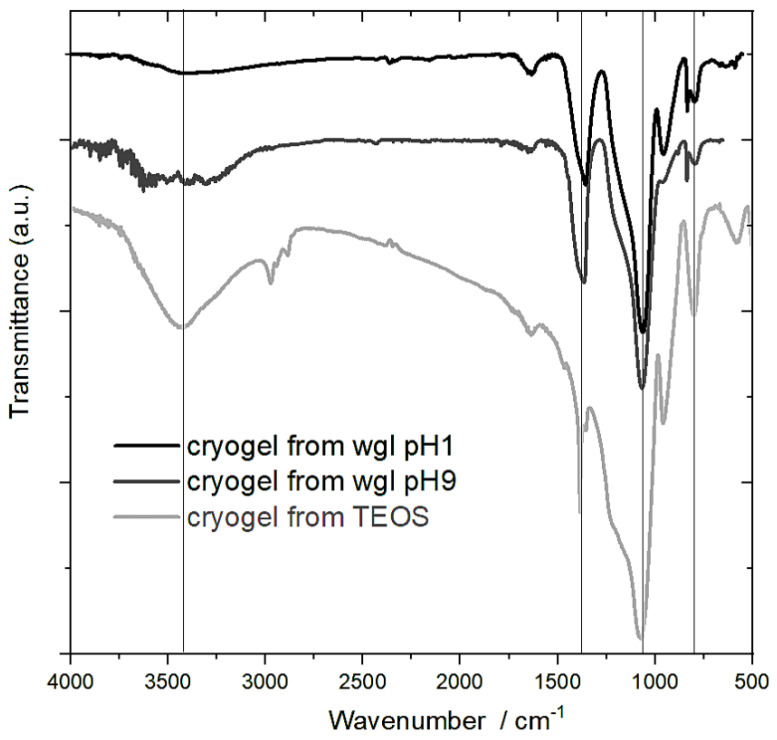
FTIR spectra of cryogels prepared from water glass in medium of pH 1 and pH 9 and from TEOS catalysed by HNO_3_.

**Figure 6 gels-08-00808-f006:**
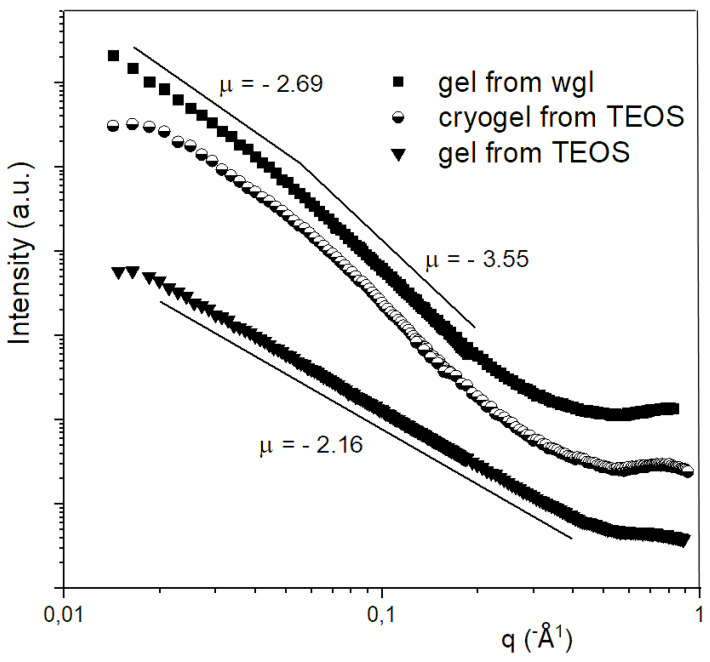
SAXS curves for alcogel from TEOS, hydrogel from water glass, and cryogel from TEOS.

**Table 1 gels-08-00808-t001:** The effect of preparation conditions on the porous structure derived from TEOS.

Molar Ratio of Solvent/Si	Catalyst, Additives	Porosity (%)	Pore Size (µm)	
			Macropores	Nanopores
10 10 *	HNO_3_	30–40 8–12	4–5 8–12	0.04–0.40 –
4	HNO_3_	35–50	4–5	0.04–0.40
4	HNO_3_ + citric acid	<50	1–2	0.03–0.15
4	NH_3_	38–50	10–15	0.50–3.00
4	HNO_3_ + propylene oxide	10–20	1.5–3.5	0.03–0.06
4	HNO_3_ + water	70–72	5–15	0.30–1.30

* The freezing rate is low: ~1 °C s^−1^. In other cases, a higher rate was used: 100 °C s^−1^.

**Table 2 gels-08-00808-t002:** The effect of pH on the gelation time and porous structure.

pH Value	HNO_3_ Solution		Gelation Time (Hours)	Pore Size (µm)	Porosity (%)
	Concentration (mol/dm^3^)	Si/HNO_3_ Molar Ratio			
11	2	11.87	20 ± 2	8–12	48–50
9 9	2 2	5.28 5.28 (HCl)	instant instant	0.15–0.18 1.5–6.0	55–60 52–58
6 6	Ion exchange 2	4.22	20 ± 2 instant	9–15	55–60
3	2 4	3.17 3.17	21 ± 2 instant	3–10 * 0.04–0.30	61–62 61–65
3					
1 1	4 6 8	1.58 1.58 1.58	6 ± 1 4 ± 1 instant	3–6 * 0.02–0.06	68–78 70–78
1					

* Hierarchical porous structure.

**Table 3 gels-08-00808-t003:** Si MAS NMR spectroscopy data.

Silica Samples	Q^2^ Peak		Q^3^ Peak		Q^4^ Peak	
	Position (ppm)	Intensity (%)	Position (ppm)	Intensity (%)	Position (ppm)	Intensity (%)
from TEOS	−91.3	8.0 ± 1	−101.8	45.4 ± 2	−112.2	46.6 ± 3
from water glass	−92.7	11.0 ± 1	−102.1	51.0 ± 3	−112.2	38.0 ± 2
cryogel from TEOS	−93.1	2.0 ± 0.3	−102.7	56.0 ± 3	−111.9	42.0 ± 2
cryogel from wgl	−93.9	1.2 ± 0.3	−103.5	59.8 ± 3	−112.7	39.0 ± 2

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
