# Peer review of "Hierarchical Porous SiO2 Cryogel via Sol-Gel Process"

_gels, 2022, doi:10.3390/gels8120808_

Round 1
Reviewer 1 Report
This is a very instructive manuscript on the optimized synthesis of inexpensive silica cryogels. The technical quality of the study is good, but the novelty is severely limited. Improvements need to be implemented before publication.
Specific comments:
1.) The recipes in the ”Experimental part” are not detailed enough to allow the reproduction of the materials. The exact amounts together with the ratios of the reagents should be given. The scale of each synthetic step has to be specified. For each step, all the process parameters should be given. A table should be presented showing both the compositions and the final characteristics of the all the synthesized materials.
2.) A key factor determining the morphology of the final materials is the freezing method, and the process parameters of the freeze-drying. These should be given in details step-by-step in order to elucidate the connection between the process parameters and the final materials characteristic.
3.) Section 2.2: The large ratio of the Q3 units are mentioned, but not explained in details. What is the mechanistic explanation in view of the preparation method for the generally large ratio, and the variation of the ratio of the Q3 units?
4.) Are the SAXS curves fitted using the Freltoft expression? The SAXS curves should be evaluated by fitting an appropriate physics-informed model to obtain the most information on morphology.
Author Response
Answers to referee:
- The scale of synthetic steps and the process parameters have been given or completed in Section 4.1.1 and 4.1.2. The effect of most important preparation condition (molar ratios) on the porous structure derived from TEOS is summarised in Table 1. In the case of using water glass, Table 2 represents the effect of the most important preparation condition (pH) on the porous structure. Showing the effect of many not so significant parameters on the materials in one table would be too chaotic.
- The details of lyophilisation conditions have been given in the section of 4.1.1 and 4.1.2. The effect of two important conditions (the rate of freezing and the quality of solvent) is represented in the Table 1 and in the text of 2.2. Section.
- In the Section of 2.2, the explanation of large ratio of Q3 has been inserted.
- The evaluation of SAXS curves has been completed by details and explanations in Section of 2.2.
Reviewer 2 Report
Dear Authors, The manuscript successfully presents the development of a new, low cost- and energy-consumption preparation route for highly porous silica systems with the gels dried under freeze drying conditions.
The obtained gels were also properly characterized.
Please, I will wish to ad only a few remarks for tiny corrections as follow:
Please rephrase the sentence from raw 71 (Starting from TEOS alcogels, from water glass hydrogels will be formed) in order to be clearly understandable that Starting from TEOS, the alcogels are formed; and from starting from water glass then the hydrogels will be formed.
Please add a visible scale bar on each SEM image.
Correct in Figure 5, Transmittance
Author Response
Answers to referee:
The sentence from row 71 has been improved according to the referee’ requirement.
The SEM images have been provided by visible scale.
The legend of Transzmittance has been corrected to Transmittance.